# Human Neutrophils Generate Extracellular Vesicles That Modulate Their Functional Responses

**DOI:** 10.3390/cells12010136

**Published:** 2022-12-29

**Authors:** María José Hurtado Gutiérrez, Frédérick L. Allard, Hugo Tshivuadi Mosha, Claire M. Dubois, Patrick P. McDonald

**Affiliations:** 1Department of Immunology and Cell Biology, Medicine Faculty, Université de Sherbrooke, CRCHUS, Sherbrooke, QC J1H5N4, Canada; 2Pulmonary Division, Medicine Faculty, Université de Sherbrooke, CRCHUS, Sherbrooke, QC J1K2R1, Canada

**Keywords:** neutrophils, extracellular vesicles, apoptosis, chemokines, NETs, autocrine

## Abstract

Neutrophils influence innate and adaptive immunity by releasing various cytokines and chemokines, by generating neutrophil extracellular traps (NETs), and by modulating their own survival. Neutrophils also produce extracellular vesicles (EVs) termed ectosomes, which influence the function of other immune cells. Here, we studied neutrophil-derived ectosomes (NDEs) and whether they can modulate autologous neutrophil responses. We first characterized EV production by neutrophils, following MISEV 2018 guidelines to facilitate comparisons with other studies. We found that such EVs are principally NDEs, that they are rapidly released in response to several (but not all) physiological stimuli, and that a number of signaling pathways are involved in the induction of this response. When co-incubated with autologous neutrophils, NDE constituents were rapidly incorporated into recipient cells and this triggered and/or modulated neutrophil responses. The pro-survival effect of GM-CSF, G-CSF, IFNγ, and dexamethasone was reversed; CXCL8 and NET formation were induced in otherwise unstimulated neutrophils; the induction of inflammatory chemokines by TNFα was modulated depending on the activation state of the NDEs’ parent cells; and inducible NET generation was attenuated. Our data show that NDE generation modulates neutrophil responses in an autocrine and paracrine manner, and indicate that this probably represents an important aspect of how neutrophils shape their environment and cellular interactions.

## 1. Introduction

Neutrophils are the most abundant circulating leukocytes. They massively infiltrate inflammatory sites where they exert various anti-microbial functions. The latter include their phagocytic capacity, which was described over a century ago by Ilya Metchnikoff [1]; their long-known ability to degranulate and produce reactive oxygen species; and their more recently discovered propensity to form neutrophil extracellular traps (NETs) [2]. This notwithstanding, research conducted over the past three decades has significantly broadened the traditional view of neutrophils as mere phagocytes that assail micro-organisms. For instance, it is now known that neutrophils respond to their environment by modulating their transcriptional capacity, and that the ensuing transcriptional programs show remarkable plasticity [3,4]. This allows neutrophils to transcribe and release a vast array of pro- and anti-inflammatory cytokines, chemokines and angiogenic factors [5], through which they influence both innate and adaptive immunity [6]. Thus, neutrophils have emerged as important regulators of varied immune responses, including T cell suppression [7,8], tissue repair [9], non-classical antigen presentation [10,11,12,13,14], and regulation of the phenotype and function of both macrophages [15,16,17] and dendritic cells [18]. 

A less studied function of neutrophils is their ability to release extracellular vesicles (EVs). These can be classified according to their origin and size into (i) apoptotic bodies, which are derived from apoptosis-triggered cell budding, with sizes ranging between 50–5000 nm; (ii) ectosomes, formed by budding and pinching of the cell membrane from viable cells, and measuring between 100–1000 nm; and (iii) exosomes, which are formed within the multivesicular bodies of the cells and typically measure less than 150 nm [19,20]. In neutrophils, EVs appear to mainly consist of ectosomes [21,22,23], though exosome formation has also been reported [24,25]. During ectosome biogenesis, cell membrane budding is driven by cytoskeletal rearrangements that entail a clustering of several surface proteins and a loss of the asymmetrical distribution of phospholipid classes between membrane leaflets [26]. As a result, ectosomes contain many of the elements present in the parent cell’s cytoplasm, and their outer membrane is enriched in several surface receptors but also features phosphatidylserine [21,22,27]. Neutrophil-derived ectosomes (NDEs) also appear to differ depending on the activation state of the parent cells. It was indeed reported that NDEs from unstimulated neutrophils do not exert the antibacterial effects observed in NDEs produced after cell activation with opsonized zymosan [23]. Similarly, NDEs from either resting or stimulated cells (PMA, fMLP and *M. tuberculosis* infection) feature a different cargo of TLR4/6 ligands, and thus, exert different effects towards autologous macrophages [28]. In addition to macrophages, other cell types have been reported to be influenced by NDEs. For instance, NDEs from fMLP-stimulated cells were found to alter the expression of proteins related to tight-junction maintenance in brain microvascular endothelial cells [29]. Likewise, dendritic cells (DCs) pre-treated with NDEs showed a downregulation of their maturation markers and induced a Th2 cytokine release profile when co-incubated with T CD4+ cells [18]. 

In view of the accumulating evidence that neutrophils can regulate the function of other cell types through the release of NDEs, we examined whether NDEs might also affect neutrophils themselves. In this study, we thoroughly characterized NDEs, as well as their production, and the signaling pathways controlling this response. We also show that NDE content quickly transfers to neutrophils, and that accordingly, NDEs can modulate neutrophil apoptosis, inflammatory chemokine generation, and NET formation. 

## 2. Materials and Methods

### 2.1. Antibodies and Reagents

Anti-human CD66b, Annexin-V-Alexa647 and Annexin-V-FITC were purchased from BioLegend (San Diego, CA, USA). Calcein Blue AM was from Invitrogen (Waltham, MA, USA). Antibodies against MPO were from Dako (A0398). Ficoll-Paque Plus was from GE Biosciences (Baie d’Urfé, QC, Canada); endotoxin-free (<2 pg/mL) RPMI 1640 was from Wisent (St-Bruno, QC, Canada). Recombinant human cytokines were from R&D Systems (Minneapolis, MN, USA). N-formyl-methionyl-phenylalanine (fMLP) and phenylmethanesulphonyl fluoride (PMSF) were from Sigma (St. Louis, MO, USA). All inhibitors, antagonists, and fluorescent probes were purchased through Cedarlane Labs (Mississauga, ON, Canada). PlaNET reagents (fluorescent chromatin-binding polymers) are no longer available from Immune Biosolutions or other suppliers; we therefore employed a close equivalent: fluorescent, 50-nm carboxylate microspheres (# 16661-10) from Polysciences Inc. (Warrington, PA, USA). These microspheres are referred to as PlaNET reagents throughout this study, since the name is neither registered as a trademark, nor under copyright. All other reagents were of the highest available grade, and all buffers and solutions were prepared using pyrogen-free, clinical grade water. 

### 2.2. Cell Isolation and Culture

Neutrophils were isolated from the peripheral blood of healthy donors under a protocol approved by an institutional ethics committee. All subjects gave written informed consent in accordance with the Declaration of Helsinki. Briefly, whole blood was collected using an anticoagulant (citrate phosphate), and successively submitted to dextran sedimentation, Ficoll separation, and water lysis—as described previously [30]. The entire procedure was carried out at room temperature and under endotoxin-free conditions. Purified neutrophils were resuspended in RPMI 1640 supplemented with 10% autologous serum, at a final concentration of 5 × 10^6^ cells/mL (unless otherwise stated). As determined by Wright–Giemsa staining and FACS analysis, the final neutrophil suspensions contained fewer than 0.1% monocytes or lymphocytes; neutrophil viability exceeded 98% after 4 h in culture, as determined by trypan blue exclusion and by Annexin V/propidium iodide FACS analysis. 

### 2.3. Isolation of Naturally Produced Ectosomes

Neutrophils (5 × 10^6^ cells/mL), resuspended in RPMI 1640 without serum, were incubated at 37 °C in 12-well plates (1 mL/well), with or without stimuli or inhibitors, in a culture incubator under a humidified 5% CO_2_ atmosphere for the indicated times. Culture supernatants were collected and centrifuged (1000× *g*, 10 min, 4 °C) to pellet any intact cells, and supernatants were further centrifuged (18,000× *g*, 15 min, 4 °C). The resulting pellets were resuspended in 1 mL PBS for each ml of starting neutrophil suspension. In some experiments, the 18,000 supernatants were further centrifuged (100,000× *g*, 60 min, 4 °C in a Beckman Optima Max ultracentrifuge using a TLA120.2 rotor) and the resulting pellets were resuspended in 1 mL PBS for each ml of starting neutrophil suspension.

### 2.4. Isolation of Ectosomes Produced by Nitrogen Cavitation

Neutrophils were suspended in RPMI 1640 to 5 × 10^7^ cells/mL, placed in 13-mL round bottom tubes on ice within a pre-cooled cavitation chamber (Parr Instrument Co., Moline, IL, USA), and pressurized under a nitrogen atmosphere (350 psi, 10 min, with constant magnetic agitation). Cavitates were collected dropwise in 50 mL polypropylene tubes and centrifuged (1000× *g*, 10 min, 4 °C) to pellet nuclei and large cell debris. The supernatants were then centrifuged (4000× *g*, 15 min, 4 °C) and the resulting pellets, containing the ectosomes, were resuspended in 1 mL PBS for each ml of starting neutrophil suspension. 

### 2.5. Ectosome Analysis by Flow Cytometry

Cavitation-derived ectosome preparations were diluted 1:10 in PBS before analysis, while naturally released ectosomes were analyzed without dilution. From these suspensions, 10 µL were mixed with 40 µL of 6 µM calcein and incubated (20 min, RT, in the dark), after which 50 µL of Annexin Binding Buffer (100 mM HEPES pH 7.4, 1.4 M NaCl, 25 mM CaCl_2_, filtrated through 0.2 µm membranes) containing 0.3 µg anti-CD66b-PE and 0.15 µg Annexin-V-Alexa647 were added. The samples were further incubated (20 min on ice, protected from light) and 100 µL ice-cold PBS were added. Samples thus treated were analyzed by FACS using a CytoFLEX S flow cytometer (Beckman Coulter); events were acquired at a low flow rate (10 µL/min) for 1 min (i.e., until the numbers stabilized) and only the events of the next 1 min were used for further analysis in the DAPI, PE and APC channels. A rinsing step consisting of acquiring a clean buffer (same flow rate and time) was always performed between samples. Data was finally analyzed using CytExpert software (version 2.4, Beckman Coulter, Indianapolis, IN, USA). 

### 2.6. Ectosome Analysis by Transmission Electron Microscopy

We followed the staining protocol described as “Protocol A” by Rikkert et al. [31]. Briefly, 10 µL of sample were placed on a 400 mesh copper grid with a carbon-coated formvar film, incubated (2 min, room temperature) and blotted before being placed for a few seconds on 10 µL of 2% uranyl acetate. Samples were then analyzed in a Hitachi H-7500 transmission electron microscope. 

### 2.7. Transfer of Vesicular Components from NDEs to Neutrophils

NDEs where loaded with 1 µL/mL Cytotrack Red or 5 µM DiO by adding a 2X solution to the same volume of NDE suspension, and incubated for 15 min (in the dark, at RT); the sample was then centrifuged (18,000× *g*, 15 min, 4 °C). The NDE pellet was washed twice and resuspended in RPMI to achieve 2 × 10^6^ NDEs/50 µL. For co-incubations, 4 × 10^5^ neutrophils were deposited in 96-well flat-bottom plates and allowed to rest (15 min, 37 °C); 50 μL of pre-warmed labeled NDE suspension was then added and the mixture was further incubated (37 °C, in the dark). At the desired timepoints, the wells’ contents were transferred to tubes containing an equivalent volume of ice-cold PBS, prior to centrifugation (500× *g*, 5 min, 4 °C). Cell pellets were resuspended in 150 µL PBS and analyzed by flow cytometry. 

### 2.8. Determination of Neutrophil Apoptosis

After the desired culture period, neutrophils (5 × 10^5^ cells) were washed twice in ice-cold PBS containing 5 mM EDTA, then once more in cold PBS, and incubated (15 min, on ice, in the dark) with FITC-conjugated Annexin V in Annexin Binding Buffer. Cells were then counterstained with propidium iodide and analyzed (minimum of 10,000 cells) on a CytoFLEX S instrument (Beckman Coulter) using the CytExpert software (v 2.4, Beckman Coulter, Indianapolis, IN, USA). 

### 2.9. ELISA Analyses

Neutrophils (3 × 10^6^ cells/600 µL) were cultured in 24-well plates at 37 °C under a 5% CO_2_ atmosphere, in the presence or absence of stimuli and/or inhibitors, for the indicated times. Culture supernatants were collected, snap-frozen in liquid nitrogen, and stored at −70 °C. Samples were analyzed in ELISA using commercially available capture and detection antibody pairs (R&D Systems). 

### 2.10. NET Microscopic Assays

For each condition, 500 µL of a neutrophil suspension (2 × 10^6^/mL in RPMI 1640/2% autologous serum) was deposited onto coverslips that had been freshly coated with poly-L-lysine and placed inside the wells of a 24-well plate; the cells were then left to adhere for 60 min in a cell culture incubator. Cells were gently washed with pre-warmed culture medium and covered with 500 µL of fresh, pre-warmed medium. Inhibitors and/or stimuli were then added, and the final volume brought to 550 µL, prior to a 4 h incubation (37 °C, 5% CO_2_). Reactions were stopped by adding 500 µL ice-cold PBS containing 1 mM PMSF, and the coverslips were placed on ice for 10 min. At this point, one of two procedures were followed.

When anti-myeloperoxidase antibodies were used for NET detection, we followed the procedure that we previously described [32]. When PlaNET reagents were used for NET detection, the liquid on the coverslips was discarded and the cells were incubated (90 min on ice, with gentle shaking) in 1 mL of cold PBS containing 1 mM PMSF and 0.2 µL PlaNET reagent. Cells were finally fixed (15 min, room temperature) in PBS containing 2% paraformaldehyde, as well as a nuclear stain. The fixed cells were then washed with PBS, and the coverslips mounted onto glass slides using a drop of mounting medium (ProLong Gold, Life Technologies), prior to epifluorescence microscopy analysis. For quantitation, 3 fields at 10× magnification were typically counted, never including the coverslip edges: this amounts to counting about 1000 neutrophils per coverslip, or about 2000 cells per experimental condition since the experiments were conducted in duplicate. Fluorescence quantitation was performed using a Java plug-in for ImageJ, which we developed (available at http://mcdonaldlab.ca/java-plug-in.html, access on 24 December 2022). 

### 2.11. Statistical Analyses

All data are represented as mean ± SEM. Unless otherwise stated, statistical differences between the two matched experimental conditions were analyzed by Student’s t test for paired data, using Prism 9 software (GraphPad, San Diego, CA, USA); data distribution passed the Shapiro–Wilk normality test.

## 3. Results

### 3.1. Isolation and Characterization of NDEs

We employed several parameters to characterize neutrophil ectosomes by flow cytometry, following MISEV 2018 guidelines; our flow and gating strategy is summarized in Figure 1. First, we found that starting to acquire data only when fluctuations in the cell and event numbers have stabilized (typically after 1 min) significantly attenuates background noise due to very small events detected by a high-resolution flow cytometer, and therefore, yields optimal reproducibility (Figure 1A). Similar observations were made when EV suspensions were spiked with fluorescent calibration beads (Appendix A). We also used these cytometry calibration beads (measuring 200, 500 and 760 nm) to identify a zone in which the ectosomes are expected to localize (Figure 1B). We next applied this gate to the analysis of EV suspensions stained with Calcein Blue AM (Figure 1C); this, in turn, allowed us to set a calcein gate (Figure 1D). EVs featuring calcein fluorescence represent closed vesicles [33]; control experiments confirmed that treating such vesicles with 0.5% NP40 for 1 min prior to FACS analysis caused a loss of their calcein fluorescence (Appendix A). Over 80% of the calcein-positive events also stained positively for surface PS (using annexin V) and for the neutrophil surface marker, CD66b (Figure 1E). We finally ensured that EV abundance, which can be quite variable depending on the experimental conditions, does not affect their detection. For this purpose, we performed serial dilutions on triple-labeled EVs prior to flow cytometry analysis. As shown in Appendix A, EV detection was tightly correlated with dilution (R^2^ > 0.998), which confirmed that our cytometry analyses are reliable across a broad range of EV concentrations. 

Because of their size and properties, we considered such triple-labeled EVs as NDEs. Consistent with this conclusion is that neutrophil pretreatment with GW4869, which inhibits nSmases and thus the release of smaller exosomes [34], had no significant effect towards NDE release (Figure 2A) or size (Figure 2B). Likewise, electron microscopy analysis of neutrophil EVs revealed closed vesicles of the expected size and shape (Figure 2C), corroborating our flow cytometry data. Finally, NDEs appear to represent the major type of EVs produced by neutrophils since ultracentrifugation of the 18,000 g supernatants (corresponding to the 18,000 g pellets from which we isolated NDEs) to 100,000 g yielded few remaining calcein-positive EVs (Figure 2D), as well as little protein content (Figure 2E). 

### 3.2. Induction of NDE Release from Human Neutrophils and Upstream Signaling Pathways

We next investigated how various physiological stimuli might influence the ability of neutrophils to generate ectosomes. As shown in Figure 3A, unstimulated neutrophils released low but detectable amounts of ectosomes, whereas exposure to LPS, TNFα, or fMLP resulted in a robust increase in ectosome production, relative to resting cells. Figure 3B shows that this induction was near maximal within 30 min and was sustained for at least 2 h, slowly decreasing afterwards; by comparison, NDE release never exceeded baseline levels in unstimulated neutrophils. Not all stimuli promoted NDE release, however. As shown in Figure 3A (lower panel), neutrophil exposure to GM-CSF, G-CSF, IFNγ or dexamethasone resulted in little or no NDE production over basal levels. 

Our previous work has demonstrated the involvement of several signaling intermediates (e.g., TAK1, p38 MAPK, MEK/ERK, Syk/Src, PI3K/Akt) in the regulation of such neutrophil functional responses as delayed apoptosis, cytokine generation, and NET formation [32,35,36,37,38,39,40,41,42,43,44]. To determine whether some of these signaling molecules also influence inducible ectosome release by neutrophils, we pretreated cells with selective inhibitors prior to stimulation with fMLP. As shown in Figure 3C, all inhibitors used (with the exception of the JNK inhibitor) significantly decreased ectosome release, albeit partially. Under no conditions were we able to completely block NDE release. 

### 3.3. Effect of NDEs on Neutrophil Functional Responses

Because NDEs were reported to affect the function of various cell types [18,28,29], we investigated whether they might similarly modulate the responses of neutrophils themselves. In this regard, an important limitation is that the amount of NDEs generated by cultured neutrophils is modest, to the point of being scarce in resting cells (Figure 2A), thus compromising studies in which NDEs from resting cells would be compared to those of activated cells. To circumvent this difficulty, we resorted to generating EVs from cavitated cells, as this reportedly yields large quantities of EVs with similar properties to those of cultured neutrophils or differentiated HL-60 cells [45]. We additionally employed a differential centrifugation approach to purify cavitation-made NDEs, since this technique yields intact organelles, including cytoplasmic granules that pellet at 15,000 g and above. As shown in Figure 4A, centrifuging cavitation-derived, post-1000 g supernatants (that are free of unbroken cells and intact nuclei) at 4000 g results in pellets containing the vast majority (about 80%) of triple-labeled NDEs, whereas a further 18,000× *g* centrifugation only pellets about 15% more NDEs (Figure 4A) but much protein content (Figure 3B), reflecting the presence of cytoplasmic granules. There remains few NDEs (Figure 4A) or total protein (Figure 4B) when the 18,000× *g* supernatants are further centrifuged at 100,000× *g*. Thus, in all subsequent experiments, we submitted neutrophil cavitates to successive 1000× *g* and 4000× *g* centrifugation steps to purify the NDEs. A direct comparison between cavitated NDEs and those released by neutrophils in culture supernatants confirmed that the former are, indeed, much more abundant, irrespective of the parent cells’ activation state (Figure 4C). Again, electron microscopy analysis of cavitated NDEs revealed closed vesicles of the expected size and shape (Figure 4D). Thus, cavitated NDEs appear to share many properties in common with those originating from cultured neutrophils, and represent an abundant source of NDEs for various uses. 

To determine whether NDEs affect neutrophil responses, we first investigated whether NDE components are incorporated by autologous neutrophils. Cells were incubated in the presence of NDEs pre-stained with Cytotrack Red and the acquisition of fluorescence by the cells was monitored by flow cytometry. Preliminary studies revealed that while a 5:1 ratio (NDEs:neutrophils) yielded a good transfer of NDE material into recipient cells, lower ratios were less efficient (not shown). As shown in Figure 5A, recipient neutrophils rapidly accumulated Cytotrack Red fluorescence from the NDEs (at a 5:1 ratio of NDEs:neutrophil); a similar effect was observed using NDEs pre-stained with the membrane dye, DiO (Appendix A), or when NDEs from cultured neutrophils (as opposed to NDEs from cavitates) were used. Moreover, when NDEs stained with Cytotrack Red were exposed to Annexin V or to proteinase K prior to co-culture with autologous neutrophils, we observed a much decreased transfer of fluorescence to recipient neutrophils (Appendix A). This indicates that both phosphatidylserine and surface proteins on NDEs play a role in their incorporation into recipient neutrophils. 

Because both membrane and intravesicular NDE components rapidly and efficiently transfer to neutrophils, we next examined whether such interactions might modulate the cells’ functional responses. In this regard, it has long been known that neutrophil exposure to physiological stimuli, such as GM-CSF, markedly delays their spontaneous apoptosis [46]. We therefore studied the effect of NDEs thereupon. As shown in Figure 5B, NDEs from cavitates did not significantly change autologous neutrophil apoptosis by themselves, whether they originated from unstimulated or fMLP-activated neutrophils. However, NDEs completely reversed the anti-apoptotic effect of GM-CSF (Figure 5B); again, a 5:1 (NDEs:neutrophil) ratio proved optimal (Appendix A). A similar outcome was observed using NDEs from cultured neutrophils (as opposed to NDEs from cavitates) (Appendix A), or when other potent pro-survival stimuli were used (Appendix A). 

Another well-characterized response of neutrophils is their ability to secrete inflammatory chemokines in response to stimuli such as LPS or cytokines [5,47,48]. As shown in Figure 5C, NDEs from cavitates modestly promoted CXCL8 release in autologous neutrophils, as did NDEs released from cultured neutrophils (Appendix A). By comparison, CCL4 was not secreted in response to NDEs under any condition tested (Figure 5C and Appendix A). However, NDEs were found to modulate the secretion of these chemokines occurring in response to the powerful inducer, TNFα. As shown in Figure 5C,D, NDEs from cavitated unstimulated neutrophils diminished TNF-elicited CXCL8 and CCL4 release. This effect was positively correlated to the NDE:neutrophil ratio (Appendix A). By contrast, NDEs from activated neutrophils (e.g., cavitates or culture supernatants from fMLP-stimulated cells) had the opposite effect and strongly enhanced chemokine secretion occurring in response to TNFα (Figure 5C,D and Appendix A). 

A more recently discovered response of neutrophils is their propension to extrude decondensed chromatin meshworks, termed NETs, which entrap various micro-organisms [2]. When we initially tested the effect of NDEs on this response, we observed that both the fMLP-elicited NET induction and even the few spontaneous NETs detected in unstimulated cells were nearly absent in cells that had been interacting with NDEs (not shown). However, control experiments revealed that this drastic effect was due to NDE interference with the NET detection reagent since NDE addition to neutrophils that had already NETed before adding PlaNET Green abolished the fluorescent signal (Appendix A); even when NDEs were added after the PlaNET reagent, an interference was still evident (Appendix A). We therefore turned to MPO staining for NET detection. However, because this approach results in a sizeable nonspecific signal [32], we conducted these analyses by treating samples with or without DNAse I at the end of the incubations, so as to isolate NET-specific fluorescence (i.e., the DNAse-sensitive signal). As shown in Figure 6, neutrophil exposure to NDEs alone promoted a NET formation comparable to that of the potent NET inducer, fMLP. However, cell pretreatment with NDEs followed by fMLP stimulation resulted in attenuated NET levels, relative to those observed in fMLP-treated cells (Figure 6B). A similar outcome was observed whether the NDEs originated from cavitates of unstimulated or activated neutrophils (Figure 6). 

## 4. Discussion

Various cell types release EVs, which help dynamically reshape their micro-environment. In the case of human neutrophils, such EVs appear to consist largely of ectosomes [21,22,23], though exosome production has also been reported [24,49,50]. In this study, we thoroughly characterized NDEs, and studied their production and the upstream signaling pathways involved. We also found that NDE content quickly transfers to neutrophils. Accordingly, we show that NDEs can modulate such core neutrophil responses as delayed apoptosis, inflammatory chemokine generation, and NET formation. 

Several methods and approaches have been used by investigators to isolate EVs (be it in neutrophils or other cell types), and there similarly exists various denominations to describe these EVs. In an attempt to standardize EV preparation and characterization, the ISEV has issued guidelines termed MISEV (Minimal Information for Studies of EVs) [20]. In the present study, we characterized the EVs isolated from human neutrophils following the MISEV 2018 standards. We mainly relied on a high resolution flow cytometry approach, using a gating strategy that focused on particles falling within the reported size for ectosomes (100–1000 nm); and several markers, including a membrane-bound protein (CD66b), an outward-facing membrane phospholipid (phosphatidylserine, detected using annexin V), intracellular proteins (Cytotrack Red), and cytoplasmic active esterases that can cleave the cell-permeable calcein-AM into a fluorescent derivative [22,51,52]. The latter parameter also ensures that the EVs we detected were closed vesicles since calcein fluorescence is otherwise lost; detergent sensitivity of triple-labeled NDEs reinforces this point. Conversely, we showed that NDEs were distinct from exosomes, as their release was unaffected by nSmase inhibition [34] and because the exosomes required much higher centrifugation speeds to pellet. Finally, electron microscopy analyses of NDEs revealed closed vesicles of the expected shape and size, corroborating our flow cytometry data. We also carefully described how NDEs were isolated, and quantified them in our various experiments. Thus, the NDEs studied herein are compliant with MISEV 2018 guidelines, which provides a reference framework for EV studies; this should help with data comparisons in future neutrophil EV studies. 

We first applied our approach to isolate and quantify NDEs, prior to studying how they are produced and what signaling pathways might control this response. In neutrophils cultured in TC-treated plasticware, low NDE levels were detected in unstimulated cells, which did not vary for up to 4 h. By contrast, neutrophil exposure to various classes of physiological stimuli (e.g., LPS, TNFα, fMLP) rapidly and robustly elicited NDE release, a strong effect being observed after only 30 min; extracellular NDE levels remained elevated for some 120 min, and decreased thereafter. These rapid EV induction kinetics agree well with those reported in previous studies in which neutrophils were stimulated with fMLP, PMA or opsonized zymosan, and EV preparations containing NDEs were analyzed [22,28,53]. Not all stimuli promoted NDE generation, however, as GM-CSF, G-CSF, IFNγ and dexamethasone proved to be ineffective inducers. We could also establish that inducible NDE generation involves several signaling molecules (e.g., TAK1, p38 MAPK, MEK/ERK, Syk, Src-related tyrosine kinases, PI3K) but not JNK. A maximal inhibition was consistently observed when blocking TAK1, in keeping with the fact that this MAP3K acts upstream of several signaling pathways in neutrophils [36,37,39,40,41,42]; differences in potency between inhibitors were, however, not found to be statistically significant by one-way ANOVA analysis. These results are consistent with the involvement of the same set of signaling intermediates in controlling other neutrophil functional responses, such as delayed apoptosis, cytokine generation, and NET formation [32,35,36,37,38,39,40,41,42,43,44]. This being said, interfering with various signaling pathways only led to a partial inhibition of NDE production, suggesting that other signaling cascades and/or cellular processes must also contribute to the phenomenon. Clearly, further studies are required to fully uncover the molecular bases of NDE generation. 

Because cultured neutrophils are inevitably exposed to their own NDEs, and because NDEs alter the function of several cell types including macrophages, DCs and endothelial cells [18,28,29], we investigated whether NDEs might affect the responses of neutrophils themselves. A first indication was that NDEs are rapidly incorporated by autologous neutrophils, with the bulk of membrane or intravesicular components having transferred by 5 min. Thus, it appears that neutrophils both secrete and re-absorb NDEs constitutively, which might explain the relatively stable extracellular NDE levels that we detected in unstimulated cells. This also indicates that the increase in extracellular NDEs observed in stimulated neutrophils could be due to enhanced NDE release, though it should be noted that re-absorption under these conditions could also be modulated. Further investigation showed that the interaction of NDEs with neutrophils does entail a modulation of their functional responses. We first established that NDEs exert pro-apoptotic effects as they largely reverse the pro-survival action of GM-CSF, G-CSF, IFNγ and dexamethasone in autologous neutrophils without significantly affecting constitutive apoptosis on their own. In view of these results, it is tempting to speculate that the notorious propensity of resting neutrophils to undergo spontaneous apoptosis might be partially related to their production of NDEs, though a method to continuously capture released NDEs would be needed to conclusively evaluate this scenario. Likewise, the relative potency of neutrophil stimuli to delay their constitutive apoptosis might be partially related to their NDE-generating ability. This would be consistent with the fact that GM-CSF, G-CSF, IFNγ and dexamethasone all potently delay neutrophil apoptosis while inducing little or no NDE release (as shown herein), whereas the comparatively modest anti-apoptotic effect of LPS or TNFα is perhaps counteracted by the ability of these stimuli to elicit NDE production. We also found that NDEs modestly induce CXCL8 (but not CCL4) secretion, in keeping with a recent study in which ectosomes from neutrophils exposed to opsonized zymosan also promoted CXCL8 production [54]; keratinocyte-derived exosomes were similarly reported to induce CXCL8 release from neutrophils [55]. In TNF-stimulated neutrophils, however, NDEs from the cavitates of unstimulated cells interfered with inducible chemokine generation, whereas NDEs from either cavitates or culture supernatants of fMLP-stimulated cells enhanced inducible chemokine secretion. Thus, while the global effect of NDEs towards the neutrophil chemokine and cytokine secretion profile would require the analysis of several more mediators, their effect on the chemokines investigated herein clearly differs depending on the activation state of the parent cells. This conclusion is in keeping with a recent report in which the authors used EVs that seemed qualitatively comparable to the NDEs used herein (though the EV numbers used were not determined). In that study, EVs from activated neutrophils also induced CXCL8 release, and this effect was diminished in the presence of EVs from unstimulated cells [54]. The same group similarly reported that the antibacterial properties of these EVs differ depending on their origin [23]. It therefore seems reasonable to contend that such divergent actions of NDEs towards neutrophil responses must reflect differences in NDE contents. In this regard, a previous study found that EVs from neutrophils (which were mainly ectosomes based on their average size) contained three times more RNA when they originated from activated cells, relative to those from unstimulated neutrophils; proteomics analyses also revealed several differences [23]. Thus, the stage is set for future studies to establish the mechanistic underpinnings of the divergent (or convergent) actions of NDEs according to the activation state of the parent neutrophils. 

The origin of the NDEs often did not matter however, as illustrated by their effects on apoptosis, or on the direct induction of CXCL8. Likewise, NDEs produced by both resting or activated neutrophils directly induced NET formation, and downregulated this response in fMLP-stimulated neutrophils. In a broader context, circulating neutrophils are typically unstimulated in healthy individuals and the few NDEs released from such cells are likely to be highly diluted, so that any impact they may have is difficult to predict. Conversely, neutrophils infiltrating inflammatory foci do so massively and have an activated phenotype. Under such conditions, NDEs being released would be expected to behave like those we isolated from activated cells, so that they would enhance the production of inflammatory chemokines by neutrophils (thereby promoting further immune cell infiltration) while perhaps limiting their survival and the extent of NET formation. All of these effects would be likely amplified since neutrophil activation leads to the production of more NDEs, as shown herein and in other studies [22,28,53]. In support of this notion, NDEs have been detected in the serum of healthy individuals, and larger numbers were detected in the serum of bacteremic patients [53]. Whatever the case might be, our data add to the existing evidence that activated neutrophils modulate their own responses and dynamically reshape their surroundings, by showing that NDE release probably contributes to these effects in response to environmental cues. 

In conclusion, the work reported herein has the potential to better focus EV research in neutrophils, by showing that such studies can be conducted in compliance with MISEV 2018 guidelines, which should greatly facilitate comparisons between various reports as the field expands. We further show that NDEs from cavitated, activated neutrophils can be used as a functional surrogate to naturally released ones, as they can be produced in large amounts, and are easy to isolate. Other applications can also be envisaged for cavitation-derived NDEs beyond their use in mechanistic studies. In this regard, a very recent report [56] showed that EVs from activated neutrophils can deliver active caspase-1 to primary tracheal epithelial cells and induce the release of IL-1α by the latter. This indeed raises the possibility of using NDEs as a therapeutic tool. 

## Figures and Tables

**Figure 1 cells-12-00136-f001:**
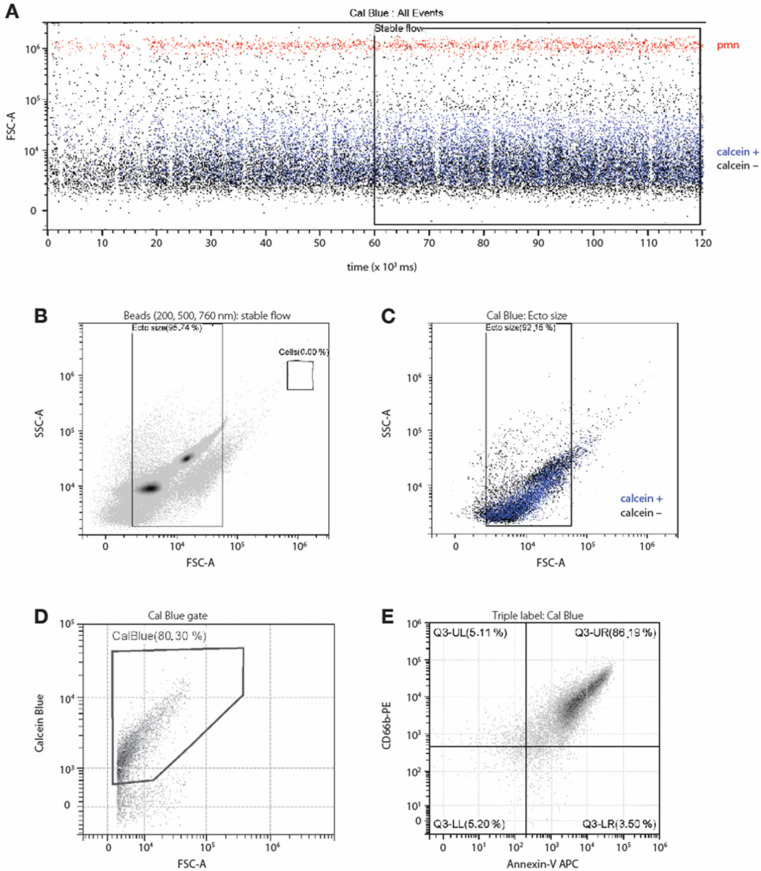
Characterization of neutrophil EVs by high resolution flow cytometry. Neutrophils were cultured in the presence of 100 nM fMLP for 30 min; culture supernatants were collected, centrifuged (1000× *g*, 10 min, 4 °C) to pellet intact cells, and the resulting supernatants were spun again (18,000× *g*, 15 min, 4 °C). The EV pellets were resuspended in 0.2 µ filtered HBSS and stained with Calcein Blue AM, prior to flow cytometry analyses. (**A**) Calcein-stained EV suspensions (blue) spiked with intact neutrophils (red) were monitored over time and data acquisitions were started after 1 min, i.e., when fluctuations in the cell and event numbers had stabilized. (**B**) Analysis of EV suspensions spiked with cytometry calibration beads (measuring 200, 500 and 760 nm) identified a region in which ectosomes are expected to localize and a size gate was established between 100 and 760 nm. (**C**) Distribution of calcein-stained EV suspensions (blue) from cultured neutrophil supernatants within this size gate. (**D**) Analysis of EVs for calcein fluorescence was used to set a gate for calcein-positive events. (**E**) Analysis of calcein-positive EVs for CD66b and Annexin-V fluorescence. Triple-positive EVs identified thusly are considered NDEs. The experiment shown in this figure is representative of NDE counts throughout the article.

**Figure 2 cells-12-00136-f002:**
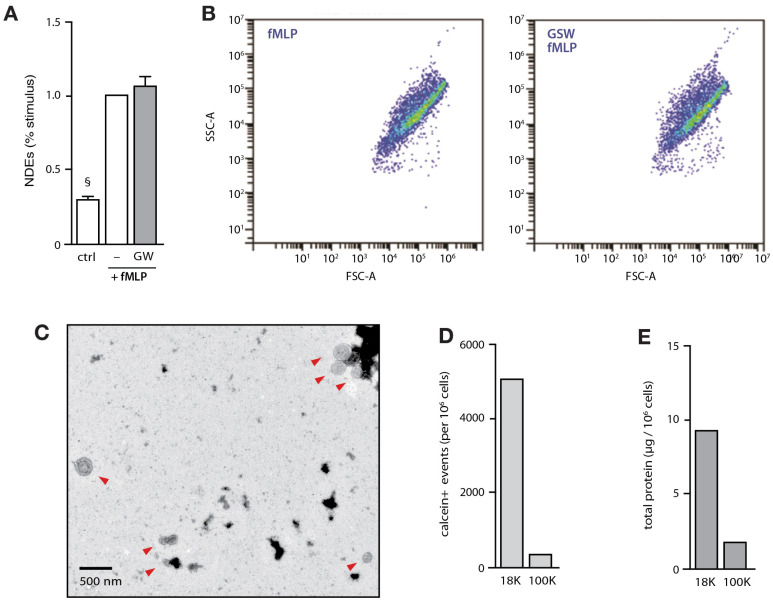
Isolation and properties of NDEs released by cultured neutrophils. (**A**) Neutrophils were cultured (15 min, 37 °C) in the presence of 20 µM GW4869 (or its diluent, 0.1% DMSO) prior to further incubation in the absence (“ctrl”) or presence of 100 nM fMLP for 120 min. Culture supernatants were collected, and NDEs were isolated and counted as described in Methods. Mean ± s.e.m. from 3 independent experiments. §, *p* < 0.01 vs fMLP-stimulated cells. (**B**) Detail of the flow cytometry analysis of the NDEs described in panel A, following the steps described in Figure 1A–C. Calcein Blue-positive events are shown. The experiment depicted is representative of three. (**C**) Transmission electron micrograph of isolated NDEs from fMLP-stimulated cells. Red arrows point to intact vesicles. The experiment depicted is representative of three. (**D**) Culture supernatants from neutrophils stimulated for 60 min with 100 nM fMLP were centrifuged at 1000 g; the resulting supernatants were centrifuged at 18,000 g and the pellets (“18K”) were processed for flow cytometry analysis of calcein-positive EVs as described in Methods. The 18,000 supernatants were further centrifuged at 100,000× *g* (2 h, 4 °C) and the resulting pellets (“100K”) processed for flow cytometry analysis of calcein-positive EVs. The experiment depicted is representative of three. (**E**) Samples depicted in panel D were also analyzed for total protein determination in Bradford assays.

**Figure 3 cells-12-00136-f003:**
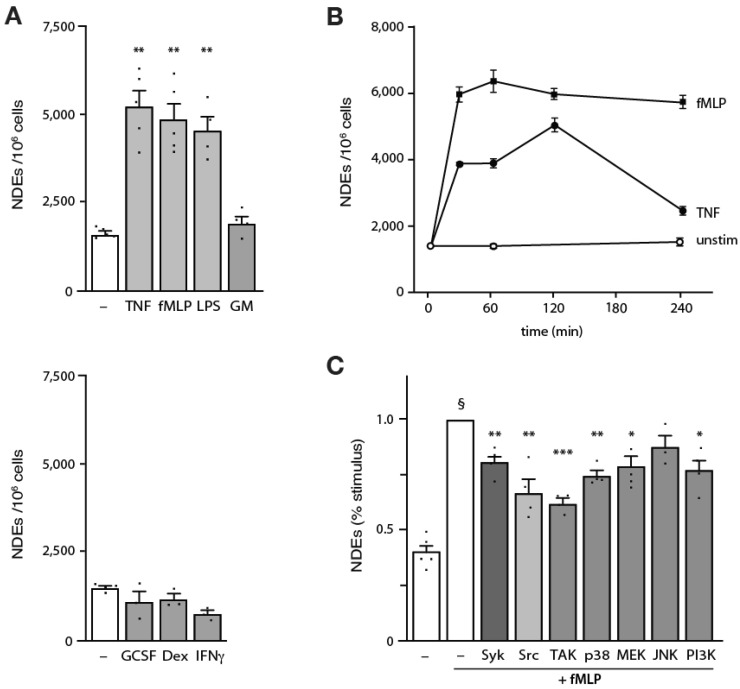
Induction of NDE generation and upstream signaling pathways. (**A**) Neutrophils were cultured for 30 min at 37 °C in the absence (“–”) or presence of 100 nM fMLP, 100 U/mL TNFα, 100 ng/mL LPS, or 1 nM GM-CSF (upper panel); or 1000 U/mL G-CSF, 100 nM dexamethasone, or 100 U/mL IFNγ (lower panel). Culture supernatants were collected, and NDEs were isolated and counted as described in *Methods*. Mean ± s.e.m. from at least 3 independent experiments. ** *p* < 0.01 vs. unstimulated cells. (**B**) Neutrophils were cultured in the absence (“unstim”) or presence of 100 nM fMLP or 100 U/mL TNFα for the indicated times. Culture supernatants were collected, and NDEs were isolated and counted as described in *Methods*. The experiment depicted is representative of three. (**C**) Neutrophils were cultured (15 min, 37 °C) in the presence of the following signaling pathway inhibitors (or their diluent, 0.1% DMSO): piceatannol (10 µM) for Syk; Src-I1 (10 µM) for Src; 5Z-7-oxozeaenol (1 µM) for TAK1; SB203580 (1 µM) for p38 MAPK; UO126 (10 µM) for MEK; SP600125 (20 µM) for JNK; and LY294002 (10 µM) for PI3K. Cells were then further incubated in the absence (“–”) or presence of 100 nM fMLP for 120 min. Culture supernatants were collected, and NDEs were isolated and counted as described in *Methods*. Mean ± s.e.m. from at least 3 independent experiments. * *p* < 0.05; ** *p* < 0.01; *** *p* < 0.001; vs. stimulus alone. §, *p* < 0.01 vs. unstimulated cells.

**Figure 4 cells-12-00136-f004:**
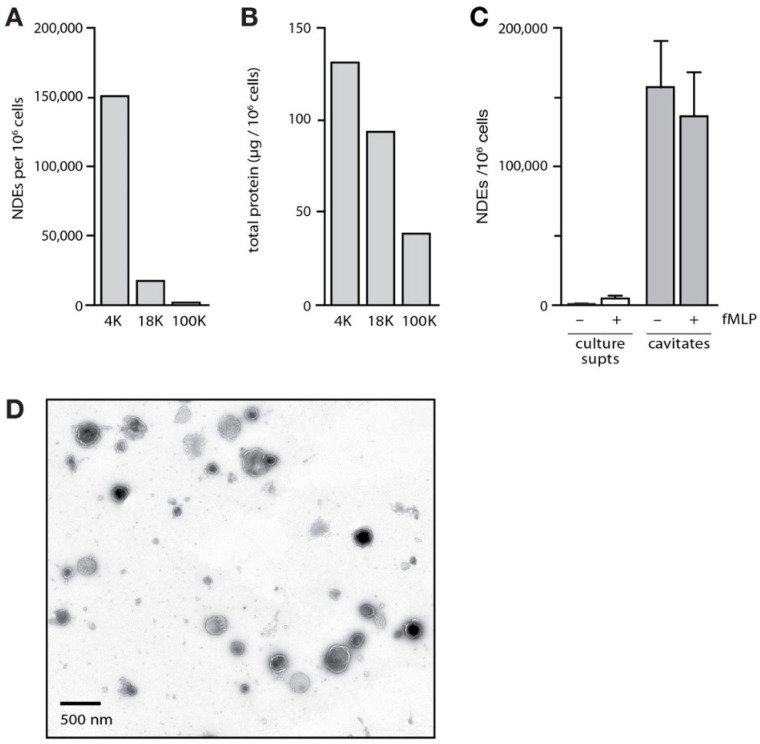
Isolation and properties of NDEs released by cavitated neutrophils. (**A**) Cavitates from neutrophils stimulated for 60 min with 100 nM fMLP were centrifuged at 1000× *g*; the resulting supernatants were centrifuged at 4000× *g* and the pellets (“4K”) were processed for flow cytometry analysis of NDEs as described in *Methods*. The post 4000 supernatants were centrifuged at 18,000× *g* and the pellets (“18K”) were processed for flow cytometry analysis of MNDEs, while the 18,000 supernatants were further centrifuged at 100,000× *g* (2 h, 4 °C) and the resulting pellets (“100K”) were processed for flow cytometry analysis of NDEs. The experiment depicted is representative of three. (**B**) Samples depicted in panel A were also analyzed for total protein determination in Bradford assays. (**C**) Neutrophils were cultured in the absence or presence of 100 nM fMLP for 60 min and NDEs were isolated from either culture supernatants or cavitates, and processed for flow cytometry analysis of NDEs as described in *Methods.* Mean ± s.e.m. from 3 independent experiments. (**D**) Transmission electron micrograph of isolated NDEs from the 4000× *g* pellets of cavitates from fMLP-stimulated cells. The micrograph is representative of three independent experiments.

**Figure 5 cells-12-00136-f005:**
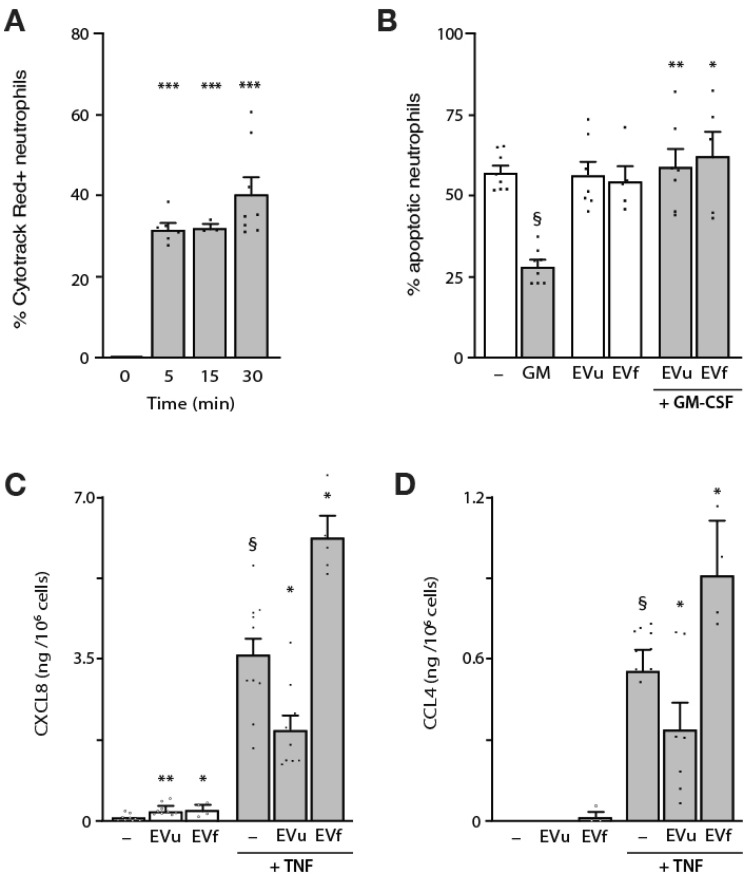
Incorporation of NDEs into autologous neutrophils and its consequence on their apoptotic rate and chemokine production. (**A**) NDEs from unstimulated neutrophils disrupted by nitrogen cavitation were stained with Cytotrack Red, washed, and co-cultured with autologous neutrophils at a ratio of 5 NDEs per recipient cell (NDEs were quantified as described in Figure 1 for triple-positive EVs). Reactions were stopped at the indicated times, and the cells were collected and analyzed by flow cytometry for Cytotrack Red fluorescence. Mean ± s.e.m. from at least 3 independent experiments. ***, *p* < 0.001; vs. unstimulated cells. (**B**) Neutrophils were cultured for 18 h in the absence of any stimulus (“–”) or in the presence of 1 nM GM-CSF, NDEs from the cavitates of unstimulated neutrophils (“EVu”, at a 5:1 NDE:cell ratio), NDEs from the cavitates of fMLP-stimulated neutrophils (“EVu”, at a 5:1 NDE:cell ratio), or a combination thereof. Cells were then processed for flow cytometry analysis of apoptotic cells (AnnexinV^+^, PI^−^). Mean ± s.e.m. from at least 5 independent experiments. *, *p* < 0.05; **, *p* < 0.01; vs. GM-CSF alone. §, *p* < 0.001 vs. unstimulated cells. (**C**) Neutrophils were cultured for 6 h in the absence of any stimulus (“–”) or in the presence of 100 U/mL TNFα, NDEs from the cavitates of unstimulated neutrophils (“EVu”, at a 5:1 NDE:cell ratio), NDEs from the cavitates of fMLP-stimulated neutrophils (“EVf”, at a 5:1 NDE:cell ratio), or a combination thereof. Culture supernatants were analyzed by ELISA for their CXCL8 content. Mean ± s.e.m. from at least 4 independent experiments, each performed in duplicate. §, *p* < 0.001 vs. unstimulated cells; *, *p* < 0.05; **, *p* < 0.01; vs. matched condition without NDEs. §, *p* < 0.001 vs. unstimulated cells. (**D**) The same samples described in panel C were analyzed by ELISA for their CCL4 content. Mean ± s.e.m. from at least 3 independent experiments, each performed in duplicate. §, *p* < 0.001 vs. unstimulated cells; *, *p* < 0.05 vs. matched condition without NDEs.

**Figure 6 cells-12-00136-f006:**
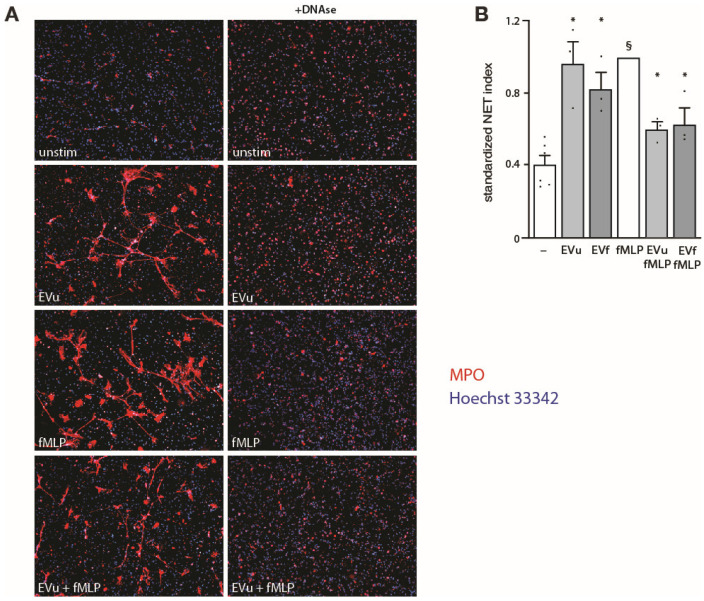
Effect of NDEs on NET production by autologous neutrophils. Cells were cultured for 4 h in the absence of any stimulus (“–”) or in the presence of 100 nM fMLP, NDEs from the cavitates of unstimulated neutrophils (“EVu”, at a 5:1 NDE:cell ratio), NDEs from the cavitates of fMLP-activated neutrophils (“EVf”, at a 5:1 NDE:cell ratio), or a combination thereof. NDEs were quantified as described in Figure 1 for triple-positive EVs. Samples were then further incubated in the presence or absence of 10 U/mL DNAse I for another 30 min to digest extracellular DNA. NET formation was assessed using MPO detection, as described in *Methods*. (**A**) A representative experiment is shown (10× magnification). (**B**) For each experimental condition, MPO fluorescence from DNAse-treated samples was subtracted from that of untreated samples so as to only quantify the NET-specific signal; these values were then standardized for the total number of nuclei. Mean ± s.e.m. of the standardized NET index from 3 independent experiments. §, *p* < 0.001 vs. unstimulated cells; *, *p* < 0.05 vs. matched condition without NDEs.

## Data Availability

The data presented in this study are available on request from the corresponding author.

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
