# Peer review of "Human Neutrophils Generate Extracellular Vesicles That Modulate Their Functional Responses"

_cells, 2022, doi:10.3390/cells12010136_

Round 1

Reviewer 1 Report

The present paper is well presented and covers an interesting topic regarding EV research. Nevertheless, some aspects of the paper need to be clarified. 

Materials and methods

The use of ultracentrifugation is stated in the results text, nevertheless, it is not covered in this section. 

Statistical analysis should include a normality test to perform a parametric test. Also, results must be presented as SD, not SEM, as it might not indicate the differences between replicates. 

Results

"followed MISEV 2018 guidelines as closely as possible"

What does this statement mean? How a methodology can be as close as possible, the guidelines are clear. You should indicate which of the parameters are being evaluated (e.g. protein quantification, particles/mL (NTA), size (DLS, NTA, TEM...), vesicles markers (flow cytometry, WB..)) as per guidelines.

"Mean ± s.e.m. from at least 3 independent experiments."

How many? Indicate the data both in bars and dispersion of points. 

Regarding NDEs effect on autologous neutrophils. Do the authors think that the NDE are internalized?

How have you determined the concentration added in the in vitro studies? 

How do the authors think that the effect of these NDE will affect other immune system cell types (e.g. macrophages, limphocytes)? 

Author Response

  1. Materials and Methods: The use of ultracentrifugation is stated in the results text, nevertheless, it is not covered in this

We have now included this information under Methods. We apologize for this oversight.

  1. Statistical analysis should include a normality test to perform a parametric test. Also, results must be presented as SD, not SEM, as it might not indicate the differences between

Data distribution passed the Shapiro-Wilk normality test in GraphPad Prism software. We have now included this information under Methods, as requested.

As for SD and SEM (i.e. SD/√n), these measures of variability are widely understood. Both are calculated for the mean of individual experiments’ means; as a result, replicates within individual experiments would not be apparent one way or the other.

  1. "followed MISEV 2018 guidelines as closely as possible" … What does this statement mean? How a methodology can be as close as possible, the guidelines are clear. You should indicate which of the parameters are being evaluated (e.g. protein quantification, particles/mL (NTA), size (DLS, NTA, TEM...), vesicles markers (flow cytometry, .) as per guideline.

In several places in the manuscript, we stated that followed MISEV 2018 guidelines, or that we are MISEV 2018-compliant. This isolated formulation ( “as closely as possible”) is a remnant from an earlier version of the manuscript, which should have been removed. It is now gone. We thank the reviewer for spotting this. As for the various parameters we used, these were already summarized in the second paragraph of the Discussion.

  1. "Mean ± s.e.m. from at least 3 independent experiments." … How many? Indicate the data both in bars and dispersion of

As requested, we updated the figures involved (Figs 3, 5,6) by showing the dispersion of data points (i.e. one point for each individual experiment), as requested. Because this now makes the graphs a tad crowded with both the datapoints and asterisks displaying statistical significance, we felt that also adding the number of experiments above the bars (when one can already count the number of datapoints) would have seriously compromised legibility.

  1. Regarding NDEs effect on autologous neutrophils. Do the authors think that the NDE are internalized? We sincerely do not know. We devised various experiments in an attempt to discriminate between mere fusion with recipient cells and actual internalization, but the outcome was never As a result, we

abandoned this line of investigation.

  1. How have you determined the concentration added in the in vitro studies?

The figure legends already specified the ratios of NDEs to recipient autologous neutrophils (5:1). NDE numbers were determined following the protocol we described for triple-labeled NDEs (Fig 1). This precision had now been added to figure legends (Figs 5 and 6).

  1. How do the authors think that the effect of these NDE will affect other immune system cell types (e.g. macrophages, limphocytes)?

Such experiments are currently being carried out for autologous PBMCs (both lymphocytes and monocytes), as well as other cell types. However, we feel that they lie beyond the scope of the present study, which aimed to characterize NDEs and their effect on autologous neutrophils.

Reviewer 2 Report

Gutiérrez and colleagues have presented an interesting manuscript showing that neutrophils can release extracellular vesicles able to modulate neutrophil responses, such as NET formation, cytokine release and apoptosis delay, in an autocrine/paracrine manner.

In general, the study is interesting. However, the work is held back by minor issues and the article might benefit when the following matters are dealt with

General points

1.    Since NDEs can be labeled, would a fluorescence microscopy estimation of size and shape be possible, if only to confirm the EM data?

2.    The authors speculate in the Discussion about potent anti-apoptotic stimuli not making EVs, but only provide GM-CSF as an example. To expand on this idea, it would be desirable to analyze other such stimuli. 

3.    The authors stress the notion that EVs from activated cells are more likely to be the biologically relevant ones, yet in Figure 6 they show only EVs from unstimulated cells. It would be desirable to show more than a single experiment using EVs from activated cells, as this would strengthen the data.

4.    The statistics used are Student’s t-test to compare matched pairs, which is fine. But in some cases, comparing various conditions with each other would be relevant, using ANOVA. For instance, in Fig 3C, how do various or inhibitors compare with each other? This would be relevant since the authors mention that TAK1 inhibitor works better than MAPK inhibitors. Is this difference significant?

Minor points (typing mistakes):

1.    Line 183 “parafornaldehyde” should read paraformaldehyde.

2.    Line 246 “following the steps described in Fig 1A, 1B, 1c”

Author Response

Since NDEs can be labeled, would a fluorescence microscopy estimation of size and shape be possible, if only to confirm the EM data?

We managed to detect fluorescent dots of roughly 0.3-0.5 µm in size at 40X magnification on a confocal microscope when imaging Calcein Blue-stained NDEs adherent to poly-L-lysine coated coverslips. However, size estimations are not reliable by this approach. Indeed, MISEV 2018 guidelines instead recommend either super-resolution microscopy (to which we did not have access) or electron microscopy (EM). In this regard, our EM data is unequivocal and already corroborates our high resolution flow data.

The authors speculate in the Discussion about potent anti-apoptotic stimuli not making EVs, but only provide GM-CSF as an example. To expand on this idea, it would be desirable to analyze other such stimuli.

We agree. We have now conducted experiments using other potent anti-apoptotic agents (e.g. G-CSF, IFNγ) and found that like GM-CSF, they do not induce NDE release (this now appears in Fig S3). Moreover, their pro-survival effect is counteracted by NDEs, again like we previously observed using GM-CSF (this now appears in Fig S5B). The relevant text has also been amended accordingly.

The authors stress the notion that EVs from activated cells are more likely to be the biologically relevant ones, yet in Figure 6 they show only EVs from unstimulated cells. It would be desirable to show more than a single experiment using EVs from activated cells, as this would strengthen the data

We have now conducted more experiments using NDEs from activated cells, which behave indeed like the ones from unstimulated cells. This new data has now been rolled into Fig 6 and the related text has been amended.

The statistics used are Student t test to compare matched pairs, which is fine. But in some cases, comparing various conditions with each other would be relevant, using ANOVA. For instance, in Figure 3C, how do various or inhibitors compare with each other? This would be relevant since the authors mention that TAK1 inhibitor works better than MAPK inhibitors. Is this difference significant?

We analyzed Fig 3C data using one-way ANOVA with Dunnett’s correction, and this did not show any statistical difference when inhibitors were compared to each other. This is now mentioned in the Discussion.

Minor points (typing mistakes)

The typos have been corrected. We thank the reviewer for spotting them.

Reviewer 3 Report

In the manuscript submitted by Hurtado Gutierrez,  et al, the authors investigate a very interesting question regarding how neutrophil derived ectosomes (NDEs) impact neutrophil function. This work contributes to an ever-growing field of understanding of the mechanisms by which neutrophils can influence both innate and adaptive immune function. The manuscript is well written and easy to follow. The majority of the figures in the manuscript characterize the process for the isolation and the validity of both naturally-produced NDEs and cavitation-derived NDEs. The authors then start to characterize the impact of their isolated NDEs on neutrophil survival, chemokine production, and NET production. While the article is asking an interesting question, there are no mechanisms given for how NDEs influence neutrophil function and this information in necessary for publication at this time. Specific points are given below:

1. For Figures 5b-d and 6a-b, there is no mechanism given for how adding NDEs impact the function that they are assessing. Is there one constituent that can be isolated from the NDEs that is responsible for this impact or is each neutrophil function differentially impacted by a different component associated with the NDEs? For example: Is this impact due to phosphatidyl serine (PS) expressed on the NDEs? Previous studies indicate that PS from ectosomes may play a role in limiting macrophage and dendritic cell activation. Experiments that utilize annexin V or a blocking agent of PS to limit the interaction of PS from the NDEs with the autologous neutrophils could test this possibility.

Author Response

  1. For Figures 5b-d and 6a-b, there is no mechanism given for how adding NDEs impact the function that they are assessing. Is there one constituent that can be isolated from the NDEs that is responsible for this impact or is each neutrophil function differentially impacted by a different component associated with the NDEs? For example: Is this impact due to phosphatidyl serine (PS) expressed on the NDEs? Previous studies indicate that PS from ectosomes may play a role in limiting macrophage and dendritic cell activation. Experiments that utilize annexin V or a blocking agent of PS to limit the interaction of PS from the NDEs with the autologous neutrophils could test this possibility.

We performed experiments in which NDEs stained with Cytotrack Red were exposed to Annexin V or to proteinase K, prior to co-culture with autologous neutrophils. Both treatments resulted in a much decreased transfer of fluorescence to recipient neutrophils, as we now show in Fig S4B. This indicates that both PS and surface proteins on NDEs play a role in their incorporation into recipient neutrophils. The text was amended accordingly.

Round 2

Reviewer 1 Report

  1. How have you determined the concentration added in the in vitro studies?

The figure legends already specified the ratios of NDEs to recipient autologous neutrophils (5:1). NDE numbers were determined following the protocol we described for triple-labeled NDEs (Fig 1). This precision had now been added to figure legends (Figs 5 and 6).

The authors have not answered the question. How did they decide which concentrations they would use? Based on previous works? Doing a dose-response evaluation? 

- The authors should clarify the significance of this work and how it can affect the field. 

- Do you think neutrophils could be modulated in vivo by NDEs as a therapeutical tool? 

Reviewer 3 Report

The authors have addressed my concerns regarding the absence of experiments that address the role of annexin V in their experiments.

Author Response

"The authors have addressed my concerns regarding the absence of experiments that address the role of annexin V in their experiments."

We did indeed, and do not know what else we should modify since the reviewer did not ask for anything more.